# Development of an Acid-Labile Ketal Linked Amphiphilic Block Copolymer Nanoparticles for pH-Triggered Release of Paclitaxel

**DOI:** 10.3390/polym13091465

**Published:** 2021-05-01

**Authors:** Svetlana Lukáš Petrova, Eliézer Jäger, Alessandro Jäger, Anita Höcherl, Rafał Konefał, Alexander Zhigunov, Ewa Pavlova, Olga Janoušková, Martin Hrubý

**Affiliations:** Institute of Macromolecular Chemistry v.v.i., Academy of Sciences of the Czech Republic, Heyrovsky Sq. 2, 162 06 Prague, Czech Republic; jager@imc.cas.cz (E.J.); hocherl@imc.cas.cz (A.H.); konefal@imc.cas.cz (R.K.); zhigunov@imc.cas.cz (A.Z.); pavlova@imc.cas.cz (E.P.); janouskova324@gmail.com (O.J.); hruby@imc.cas.cz (M.H.)

**Keywords:** MPEO-*b*-PCL nanoparticles, acyclic ketal group, paclitaxel, human HeLa carcinoma cells

## Abstract

Here, we report on the construction of biodegradable poly(ethylene oxide monomethyl ether) (MPEO)-*b*-poly(ε-caprolactone) (PCL) nanoparticles (NPs) having acid-labile (acyclic ketal group) linkage at the block junction. In the presence of acidic pH, the nanoassemblies were destabilized as a consequence of cleaving this linkage. The amphiphilic MPEO-*b*-PCL diblock copolymer self-assembled in PBS solution into regular spherical NPs. The structure of self-assemble and disassemble NPs were characterized in detail by dynamic (DLS), static (SLS) light scattering, small-angle X-ray scattering (SAXS), and transmission electron microscopy (TEM). The key of the obtained NPs is using them in a paclitaxel (PTX) delivery system and study their in vitro cytostatic activity in a cancer cell model. The acid-labile ketal linker enabled the disassembly of the NPs in a buffer simulating an acidic environment in endosomal (pH ~5.0 to ~6.0) and lysosomal (pH ~4.0 to ~5.0) cell compartments resulting in the release of paclitaxel (PTX) and formation of neutral degradation products. The in vitro cytotoxicity studies showed that the activity of the drug-loaded NPs was increased compared to the free PTX. The ability of the NPs to release the drug at the endosomal pH with concomitant high cytotoxicity makes them suitable candidates as a drug delivery system for cancer therapy.

## 1. Introduction

Over the past two decades, amphiphilic block copolymers (composed of hydrophilic and hydrophobic blocks) have been extensively studied [1,2,3,4,5]. Because of their unique capability to self-assemble in aqueous media, they attract considerable interest as potential biomedical applications in drug delivery and in the gene transfection field [6,7,8]. One key feature of these materials is associated with their capability to bear the lipophilic agent (drug) and release it in a controlled manner [7,9]. However, these nanoassemblies consist of a compact core of the insoluble block, which works as a reservoir for drugs surrounded by a flexible corona of the soluble block and often shown the slow release profile of encapsulated molecules [10,11,12,13]. It is of particular importance that hydrophilic corona provides a highly water-bound barrier to ensure colloidal stability, reduction in the rate of opsonin adhesion, and uptake by cells of the reticuloendothelial system (RES), which prolongs blood circulation lifetime [10,14,15,16].

In addition, self-assembled NPs gain much more relevance in biomedical applications, especially if they are tailored to be degradable as a response to external stimuli. Such stimulus may be the enzymatic removal of protecting groups [16], light [17], temperature [18], redox gradient [19], or change of pH [20]. The degradation through stimuli-responsive involving the cleavage of labile linkages in response to external stimuli has been widely studied as a promising platform for the enhanced/controlled release of encapsulated molecules [21,22]. To address this issue, the ability of such a system to generate a nanoparticle release of cargo at selective pH in response to the acidic tumor environment intracellularly triggered by endo/lysosomes has been extensively studied. Besides, the targeted delivery to tumors can be achieved through cleavage of these linkages at acidic tumor conditions (cancer cells and tumor tissues ranging from pH 5.7 to 7.2); the drug can be rapidly released accompanied by dissolution of nanocarriers [23,24].

Considering the high acidity in tumor tissues and intracellular compartments, several polymers responsive to pH-sensitive hydrolysis, enzymatic degradation, and/or redox reactions could be suitable for triggering drug release undergoing degradation to various extents in vitro and/or in vivo, acid-labile groups. Such acid-labile linkers, can be hydrazone [25], orthoester [26,27], imine [28], ketal, or acetal [29,30], which have often been used as a responsive group to construct pH-responsive (co)polymers [31,32]. Among all these mentioned above, acid-degradable (co)polymers that contain ketal/acetal labile linkers on the polymer backbone or as pendant groups enabling drug attachment and these NPs have received great interest due to their special features [33,34].

Among the various approaches reported to date, previously we have studied the introduction of acid-cleavable ketal linkage at the junction of poly(ε-caprolactone) (PCL) hydrophobic and poly(ethylene oxide) (PEO) hydrophilic blocks driven by the fact that could have great potential as a drug delivery system [29]. The choice of PCL and PEO as building blocks is due to their excellent biodegradability and biocompatibility. Indeed, PCL is aliphatic hydrophobic polyester; it has been approved by the food and drug administration (FDA) and widely used for biomedical applications [35,36]. Furthermore, PEO is a hydrophilic, water-soluble, and very flexible biocompatible polymer that is non-toxic and easily eliminated from the body [37].

The present study was undertaken to investigate the potential of MPEO-*b*-PCL block copolymer NPs as a tumor-specific drug delivery carrier (Figure 1). The cytotoxic drug paclitaxel (PTX) was chosen as a model hydrophobic drug to evaluate the loading and triggered release profiles of the NPs. PTX was encapsulated in the hydrophobic core by hydrophobic interactions. The drug release and cytotoxic activity of the novel NPs prepared from the MPEO_44_-*b*-PCL_17_ block copolymer were evaluated on human HeLa carcinoma cells in vitro. Due to the specific chemical structure of the block copolymer, NPs disassemble and release the drug cargo under mildly acidic conditions (which simulate the acidic environment in endosomal and lysosomal compartments), exerting in vitro cytostatic efficacy on HeLa human cervical carcinoma cell line. The hydrolysis of the ketal linkage results in neutral degradation products, which can be easily excreted, avoiding accumulation and likely inflammatory responses.

## 2. Materials and Methods

### 2.1. Materials

Chemicals were purchased from Sigma-Aldrich at the highest purity and, if not stated otherwise, were used as received.

### 2.2. Synthesis of MPEO_44_-b-PCL_17_ Diblock Copolymer Containing a Ketal Group

The synthesis and characterization of MPEO-*b*-PCL diblock copolymer are described in our previous publication [29]. Briefly, a seven-step synthetic method that combines carbodiimide chemistry (DCC method), a “click” reaction, and ring-opening polymerization (ROP) was employed to successfully produce a series of MPEO-*b*-PCL diblock copolymers (herein referred to as MPEO_44_-*b*-PCL_17_). Firstly, through a four-step pathway containing different synthetic routes, the low-molecular-weight compounds were prepared, further used as precursors for constructing the acid-labile ketal group. Then the block copolymers were synthesized in the next three steps via a DCC method, “click” reactions, and ROP (Appendix A).

### 2.3. Characterisation Techniques

^1^H NMR and ^13^C NMR spectra (300 MHz, respectively) were recorded using a Bruker Avance DPX 300 NMR spectrometer with CDCl_3_ as the solvent at 25 °C. The chemical shifts were relative to TMS using hexamethyldisiloxane (HMDSO, *δ* = 0.05 and 2.0 ppm from TMS in ^1^H NMR and ^13^C NMR spectra) as the internal standard.

The number-average molecular weights (*M*_n_), weight-average molecular weights (*M*_w_), and dispersities (*M*_w_/*M*_n_*)* of the synthesized macromer, macroinitiator, and final block copolymer were determined by size exclusion chromatography (SEC). SEC analysis was performed using an SDS 150 pump (Watrex, Carolina Centrum, Czech Republic) equipped with refractometric (Shodex RI-101, Tokyo, Japan) and UV (Watrex UVD 250, Carolina Centrum, Czech Republic) detectors. The separation system consisted of two PLgel MIXED-C columns (Polymer Laboratories, Cambridge, UK) and was calibrated with polystyrene standards (PSS, Esslingen, Germany). THF was used as the mobile phase at a flow rate of 1.0 mL·min^−1^ at 25 °C. Data collection and processing were performed using the Clarity software package.

The DLS measurements were performed using an ALV CGE laser goniometer consisting of a 22 mW HeNe linear polarized laser operating at a wavelength (λ = 632.8 nm), an ALV 6010 correlator, and a pair of avalanche photodiodes operating in pseudo-cross-correlation mode. The samples were loaded into 10 mm diameter glass cells and maintained at 37 ± 1 °C. The data were collected using the ALV Correlator Control software and the counting time was 45 s. The measured intensity correlation functions g_2_(*t*) were analyzed using the algorithm REPES (incorporated in the GENDIST program) [38], resulting in the distributions of relaxation times shown in equal area representation as *τ*A(*τ*). The mean relaxation time or relaxation frequency (*Γ* = *τ*^−1^) is related to the diffusion coefficient (*D*) of the nanoparticles as $D=\frac{\Gamma }{q^{2}}$ where $q=\frac{4\pi n\sin\frac{\theta }{2}}{\lambda }$ is the scattering vector being *n* the refractive index of the solvent and *θ* the scattering angle. The hydrodynamic radius (*R*_H_) or the distributions of *R*_H_ were calculated by using the Stokes-Einstein relation:(1)$$R_{H}=\frac{k_{B}T}{6\pi \eta D}$$
where $k_{B}$ is the Boltzmann constant, *T* the absolute temperature, and *η* the viscosity of the solvent.

In the static light scattering (SLS), the scattering angle was varied from 30° to 150° with a 10° stepwise increase. The absolute light scattering is related to weight-average molar mass (*M_w(NP)_*) and to the radius of gyration (*R*_G_) of the nanoparticles by the Zimm formalism represented as:(2)$$\frac{K_{c}}{R_{\theta }}=\frac{1}{M_{w\left(NP\right)}}\left(1+\frac{R_{Gq^{2}}^{2}}{3}\right)$$
where *K* is the optical constant, which includes the square of the refractive index increment (d*n*/d*c*), *R_θ_* is the excess normalized scattered intensity (toluene was applied as standard solvent), and *c* is the polymer concentration given in mg mL^−1^. The refractive index increment (d*n*/d*c*) of the MPEO_44_-*b*-PCL_17_ NPs in PBS (0.140 g L^−1^) was determined using a Brice–Phoenix differential refractometer operating at λ = 632.8 nm.

The average ζ-potential of the NPs was performed using a Zetasizer Nano-ZS, Model ZEN3600 Instrument (Malvern Instruments, Malvern, UK). The equipment measures the electrophoretic mobility (U_E_) and converts the value into ζ-potential (mV) through Henry’s equation (Equation (3)) where ε is the dielectric constant of the medium and *f*(ka) is Henry’s function calculated through the Smoluchowski approximation with *f*(ka) = 1.5 which was calculated using the DTS (Nano) program.
(3)$$U_{E}=\frac{2\varepsilon \zeta f\left(\mathrm{ka}\right)}{3\eta }$$

The small-angle X-ray scattering (SAXS) experiments were performed on the P12 BioSAXS beamline at the PETRA III storage ring of the Deutsche Elektronen Synchrotron (DESY, Hamburg, Germany) at 20 °C using a Pilatus 2M detector (Dectris, Baden, Switzerland) and synchrotron radiation with a wavelength of λ = 0.1 nm. The sample-detector distance was 3 m, allowing for measurements in the q-range interval from 0.11 to 4.4 nm^−1^. The q-range was calibrated using the diffraction patterns of silver behenate. Background scattering of the solvent was carefully subtracted. We have used SASFit software [39,40] and the combination of two models to describe the scattering behavior of nanoparticles. The expression for the sphere with gaussian chains attached has been derived by Pedersen and Gerstenberg [41,42]. Equation (4) for generalized Gaussian coil is as follow:(4)$$I\left(q\right)=I_{0}\left(\frac{1}{\nu U^{\frac{1}{2\nu }}}\gamma \left(\frac{1}{2\nu },U\right)-\frac{1}{\nu U^{\frac{1}{\nu }}}\gamma \left(\frac{1}{\nu },U\right)\right)$$
where $U=\left(2\nu +1\right)\left(2\nu +2\right)\frac{q^{2}R_{g}^{2}}{6}$; *ν* is the excluded volume parameter from the Flory mean-field theory.

Transmission electron microscopy (TEM) analysis was carried out on a Tecnai G2 Spirit Twin at 120 kV (FEI, Lausanne, Switzerland).

### 2.4. Nanoparticle (NP) Preparation

A preheated (40 °C) acetone solution (2.0 mL) containing the MPEO_44_-*b*-PCL_17_ block copolymer (5.0 mg) and the chemotherapeutic PTX (0.1 mg) was added drop-wise (EW-74900-00, Cole-Parmer^®^) into a pre-heated (40 °C) PBS solution (4 mL, pH ~7.4). The pre-formed NPs were allowed to self-assemble, and then the solvent was evaporated under reduced pressure. The final concentration was adjusted to 1 mg·mL^−1^ (NPs) using PBS at pH ~7.4.

### 2.5. Drug Loading and Drug Loading Efficiency

PTX loaded into the NPs was measured by HPLC (Shimadzu, Japan) using a reverse-phase column Chromolith Performance RP-18e (100 × 4.6 mm^2^, eluent water-acetonitrile with acetonitrile gradient 0–100 vol %, flow rate = 1.0 mL·min^−1^). Firstly 100 µL of the drug-loaded NPs was collected from the bulk sample, filtered (0.45 µm), and diluted to 900 µL with Acetonitrile (Lach-ner, Neratovice, Czech Republic). Such procedure led to NPs dissolution. Afterward, 20 µL of the final sample was injected through a sample loop. PTX was detected at 227 nm using ultraviolet (UV) detection. The drug-loading content (LC) and the drug-loading efficiency (LE) were calculated by using the following equations:(5)$$\mathrm{LC}\left(\%\right)=\frac{\mathrm{drug}\;\mathrm{amount}\;\mathrm{in}\;\mathrm{nanoparticles}}{\mathrm{mass}\;\mathrm{of}\;\mathrm{nanoparticles}}\times 100$$
(6)$$\mathrm{LE}\;\left(\%\right)=\frac{\mathrm{drug}\;\mathrm{amount}\;\mathrm{in}\;\mathrm{nanoparticles}}{\mathrm{drug}\;\mathrm{feeding}}\times \;100$$

### 2.6. Drug Release Experiments

The in vitro release of PTX from the block copolymer NPs was studied in pH-adjusted release media (pH ~7.4 and ~5.0) at 37 °C. Aliquots (500 µL) of drug-loaded block copolymer NPs in PBS were loaded into 36 Slide-A-Lyzer MINI dialysis microtubes with MWCO 10,000 (Pierce, Rockford, IL, USA). These microtubes were dialyzed against 4 L of pH-adjusted PBS buffer gently stirred. The drug release experiments were done in triplicate. At each sampling time, it was removed three microtubes from the dialysis system, and 300 µL from each microtube was sampled and diluted to 1.0 mL by using Acetonitrile (Lach-ner, Czech Republic). The PTX content at each sampling time was then determined via HPLC by applying the same procedure used to determine LC and LE.

### 2.7. Cell Culture and In Vitro Experiments

All HeLa cells experiments were performed according to the protocol used in our previously published paper [43]. Briefly, the HeLa cells were cultivated in Dulbecco’s Modified Eagle’s Medium (DMEM) supplemented with 10% fetal calf serum, 100 units of penicillin, and 100 µg·mL^−1^ of streptomycin (Life Technology, Waltham, MA, USA). The cells were grown in a humidified incubator at 37 °C with 5% CO_2_. For the cytotoxicity assay, 5000 cells per well were seeded in duplicates in 96 flats bottoms well plates in 100 µL of media 24 h before adding the NPs. For adding of the particles, the volume was calibrated to 80 µL, and 20 µL of the five times concentrated dilution of PTX or particle dispersion were added per well to a final PTX concentration ranging from 10^−5^ to 5 μg·mL^−1^. All dilutions were made in full incubation medium under thorough mixing of each dilution step. The sample concentrations of the PTX-loaded particles were adjusted to contain the same total amount of PTX as the samples with free PTX. The cells were incubated with the free drug or NPs for 24 h or 48 h. Then 10 µL of alamarBlue^®^ cell viability reagent (Life Technologies, Waltham, MA, USA) were added to each well and incubated a minimum for 3 h at 37 °C. The fluorescence of the reduced marker dye was read with a Synergy H1 plate reader (BioTek Instruments, Winooski, VE, USA) at excitation 570 and emission 600 nm. The fluorescence intensity of the control samples (with no drug or particles added) was set as a marker of 100% cell viability. The fluorescence signal of “0% viability samples“(where all cells were killed by the addition of hydrogen peroxide) was used as background and subtracted from all values prior to calculations. The non-toxic character of the blank particles without drug was shown by incubation of cells up to 0.67 mg·L^−1^ of blank particles. This corresponds to the amount of polymer that is contained in the samples of drug-loaded particles with 5 µg·L^−1^ total PTX content. For the cell experiments, the PTX-dilutions in incubation medium were made from a PTX stock solution of 120 µg·mL^−1^ in PBS/DMSO (96.5: 3.5 *v*/*v*) [44]. Precipitation of the hydrophobic PTX out of the cell culture medium can therefore be excluded because the PTX was previously fully dissolved in a PBS/DMSO solution (96.5: 3.5 *v*/*v*) and subsequently diluted in the serum-supplied medium under thorough mixing. Consequently, even at maximal PTX-concentration of 5 µg·mL^−1^, the final DMSO concentration in the incubation medium was below 0.2% and, therefore, no effect on cell vitality in the applied setup [42,43]. All the cell experiments were the average of at least 4 measurements (*n* ≥ 4).

## 3. Results and Discussion

A multi-synthetic pathway was developed in order to obtain well-defined acid-labile self-assemble copolymer NPs, which could release hydrophobic drugs at an increased rate at relevant mild acidic conditions. Due to careful selection of macromolecular characteristics, such as the molecular weight and the relative block length, the MPEO_44_-*b*-PCL_17_ block copolymer self-assembles in PBS (pH ~7.4), forming spherical NPs with a bulky core capable of encapsulating and controlling the release of the hydrophobic chemotherapeutic drug-PTX. This should raise potential PTX toxicity to cancer cells, while after releasing the cargo, the nanocarriers are further disassembled into environmentally neutral degradation products (Figure 1, bottom).

The chemical structure, composition, molecular weight, and dispersity of the obtained MPEO_44_-*b*-PCL_17_ block copolymer were confirmed by ^1^H and ^13^C NMR spectroscopy (see Appendix A) and SEC chromatography (Appendix A, black curve), as well. The macromolecular characteristics of the block copolymer are listed in Table 1.

The acid-responsive degradation of the MPEO_44_-*b*-PCL_17_ block copolymer linkage at pH ~5.0 for 48 h was studied in detail by SEC analysis (Appendix A red curve) and ^13^C NMR spectroscopy (Appendix A). The SEC chromatogram of the diblock copolymer before and after hydrolytic degradation, as indicated by the overlap of the SEC traces (see Appendix A, ESI), showed the full disappearance of the chromatogram due to the parent copolymer (Appendix A, black curve). Two major populations have been formatted, corresponding to the PEO and PCL homopolymer species upon the cleavage of ketal linkage at the block junctions in acidic pH, respectively (Appendix A red curve). Besides, the cleavage of the ketal group under acidic conditions in the composition of MPEO_44_-*b*-PCL_17_ diblock copolymer confirmed by ^13^C NMR spectroscopy (Appendix A), as was demonstrated in our previous paper [29]. However, the performed analyzes showed strong evidence for the high selectivity of the hydrolysis towards the ketal group linker of the diblock, and no hydrolytic degradation of the PCL backbone was observed over the period mentioned above.

After solubilization in acetone, the MPEO_44_-*b*-PCL_17_ diblock copolymer underwent nanoprecipitation and self-assembled into spherical NPs in PBS, encapsulating the PTX chemotherapeutic (see, Section 2). The SLS and DLS data revealed the assembly of well-defined NPs after acetone evaporation (Figure 2). The linear relationship resulted from the plot of the relaxation frequency (*I*) versus the square of the scattering vector (*q*^2^) (Figure 2a), which indicates the behavior of Brownian diffusion from spherical particles. Using the slope of the diffusion coefficient and employing the Stokes-Einstein equation (Equation (1), Mat. and Methods), an apparent *R*_H_ of 32.1 nm was estimated, which was in good agreement with the particle size measured at a fixed angle of 90° (Figure 2b).

From the Zimm analysis of the SLS data (see Section 2), the values of the radius of gyration (*R*_G_) were determined from the slope of the curve as being equal to 28.3 nm, and from the inverse of the intercept, the values of the NPs molecular weight (*M*_w_(NP’s)) were determined as being equal to 2.9 × 10^7^ g moL^−1^. The aggregation number of the NPs (*N*_agg_) were then calculated as *N*_agg_ = *M*_w_(NP’s)/*M*_w_(SEC) = 9265 chains being compatible with *N*_agg_ related in literature for MPEO-*b*-PCL NPs [45].

It is well established that the *R*_G_/*R*_H_ ratio may provide qualitative information related to the architecture of the self-assemblies in the solution. The obtained *R*_G_/*R*_H_ ratio was equal to 0.88, which is higher than that predicted for homogenous sphere (0.77) and was compatible with the formation of spherical block copolymer NPs with solvated shells [18,46]. The average ζ-potential of the studied self-assemble block copolymer NPs performed at pH 7.4 (PBS buffer, 0.01M) was close to neutrality (~0.5 mV) which indicates that particles were sterically stabilized by the PEO hydrophilic shell.

Thermodynamically stable polymeric NPs with a hydrodynamic diameter (2*R*_H_ = *D*_H_) of ~64 nm were obtained. These NPs were perfectly suited for drug delivery by specific accumulation in solid tumor tissue by the EPR effect. The optimal particle size for the EPR effect is usually stated to be ~20 to 70 nm [47]. Moreover, another previously mentioned valuable target in cancer therapy is the acidic environment in endosomal (pH ~5.0 to ~6.0) and lysosomal (pH ~4.0 to ~5.0) compartments. Therefore, the degradation behavior of the polymer NPs containing the acid-labile ketal group was evaluated under acidic physiological conditions (pH ~5.0; at 37 °C) using DLS, SAXS, and TEM.

The DLS data from NPs measurements over 48 h at pH ~7.4 and ~5.0 are depicted in Figure 3. The DLS data clearly demonstrated that the hydrodynamic radius (*R*_H_) of NPs remains unchanged at pH ~7.4 for 48 h (black circles, Figure 3a), whereas it continuously increased at pH ~5.0 over the 48 h (blue circles, Figure 3a). Additionally, we also observed the appearance of a scattering population that increased over time, that corresponded to smaller sizes fraction with *R*_H_ ~2 to 15 nm (blue circles, Figure 3a). A representative average intensity distribution of the NPs after 48 h at pH ~7.4 and ~5.0 are shown in Figure 3b. In contrast to the unimodal average size distribution of the NPs at pH ~7.4 (black circles, Figure 3b), the NPs average size distribution at pH ~5.0 is bimodal and shows two main population of particles with *R*_H_ ~15 nm and *R*_H_ ~58 nm (blue circles, Figure 3b), respectively.

Moreover, the changes in the NPs inner structure under degradation were also evaluated by SAXS (Figure 4). The scattering curves were modeled using a combination of core-shell and Gaussian coil model (see Section 2). The parameters extracted from the SAXS curve fittings were radius of the core (*R*), the radius of gyration of the shell (*Rg*_shell_), and the radius of gyration of the gaussian chains (*Rg*_gauss_) for the particles and random coils in the solution, respectively. The values found at pH ~7.4 were *R* = 23.4 nm and *Rg*_shell_ = 1.1 nm, resulting in a total particle radius (*R* + *Rg*_shell_) of 24.5 nm, which was only slightly smaller than the *Rg* determined through static light scattering (*R*_G_ = 28.3, Figure 2a). However, the profile of the scattering curve changes drastically after degradation at pH ~5.0 (Figure 4a). It was observed a reduction in the scattering intensity *I*(q) at the zero scattering angle *I*(0) and an increase in the scattering intensity in the high-q region (blue circles, Figure 4a), which indicated a decrease in the particle’s molecular weight and the appearance of a scattering population of free chains analogous to that observed by DLS. In this case, the values found at pH ~5.0 were *R* = 19.5 nm, *Rg*_shell_ = 5.6 nm, and *Rg*_gauss_ = 3 nm for the spherical and Gaussian coil model, respectively. The overall reduction in the dimension of the particles after incubation at pH ~5.0 was confirmed by the shift on the distribution function p(R) from the sphere model to lower values (Figure 4b). Furthermore, the increasing of *Rg*_shell_ value, as well as the appearance of the free gaussian chains in solution, indicates, respectively, the increased swelling of the NPs shell and the release of MPEG from the NPs surface.

The overall scattering changes observed confirmed the sensitivity of the NPs to degradation under acidic conditions [29] and provide important information concerning the degradation process (Table 2). The appearance of distinct populations of particles along the time at pH ~5.0 seemed to be related to a continuous disassembly and aggregation process of the NPs after the hydrolysis of the ketal group [18]. Taking into consideration that the ketal group was preferentially localized at the NPs interface, between the PCL core and the PEO shell, the hydrolysis of the ketal group released the PEO chains from the NPs surfaces, increasing the NPs hydrophobicity, which induced its disassembly and aggregation. Therefore, the results obtained by DLS showed an increase in the NPs size from ~64 to ~116 nm, as well as the appearance of the scattering population at the smaller sizes between ~4 to ~30 nm. This is probably due to disasembled/aggregated NPs and the released of free PEO chains, respectively. (Figure 3). Likewise, this was observed by SAXS by the decrease in size and scattering component that describes the NPs and the increase in concentration of the free PEO chains. Similarly, the TEM images (Figure 5) showed a comparable increase in the size of the MPEO_44_-*b*-PCL_17_ block copolymer NPs at pH ~5.0 (Figure 5b) when compared to the NPs at pH ~7.4 (Figure 5a) after 24 h, which confirms that after degradation only undefined aggregates and disassembled NPs with free MPEO chains coexisted in solution.

The release profile of the chemotherapeutic PTX was explored towards mimicking the target acidic environment in endosomal and lysosomal compartments (pH ~5.0, 37 °C)( Figure 6). Furthermore, to simulate physico-chemical conditions during transport in the blood and in normal healthy tissues, the release experiments at pH ~7.4 and 37 °C were also performed, as well (Figure 5). The obtained results suggest that the acid pH accelerates the release of the drug from the NPs, most likely due to the NPs physical destabilization (pH-triggered disassembly-aggregation, Figure 3). The drug cargo was released almost twice as efficiently (~70% released) within 48 h at pH ~5.0 (mimicking intracellular environment) than at physiological conditions of pH ~7.4. On the other hand, at pH ~7.4, only ~36% of the drug-loaded into the NPs cores was released.

The drug-release profile was considered not optimal; however, a yet faster release was expected in contact with more complex media such as the serum-supplied cell culture medium. To investigate the inhibitory effect on tumor cells, the MPEO_44_-*b*-PCL_17_ NPs were loaded with the antitumor drug PTX with an overall cargo rate of around 2.0 wt% (loading efficiency of 92%, see Section 2). Given the hydrophobicity of the NPs core and the PTX (negligible free PTX was observed), no additional purification step was carried out. The cell viability assay was used to document in vitro cytotoxicity as a classical approach to evaluate the direct effect of the drug carrier NPs on target cancer cells. The HeLa cell line was selected as a widely used and well-studied cancer cell model system [47]. The drug-loaded NPs were incubated with the HeLa cells, and the in vitro cytotoxicity after 24 h and 48 h of incubation was assessed by alamarBlue^®^ assay (Figure 7). After 48 h incubation with the cells, the PTX-loaded MPEO_44_-*b*-PCL_17_ NPs exhibited significantly stronger toxicity than the free drug (Figure 7b). In contrast, the drug-free NPs showed only negligible cytotoxicity to the cancer cells (Appendix A). This increased cytotoxicity of the drug-carrying NPs compared to the free drug was supposedly owed to endocytotic uptake [48]; at low drug concentrations (below 1 µg·mL^−1^, see Figure 7b), the endocytotic uptake of the drug-loaded nanocarriers would be more efficient than the uptake of the free drug into the cells. With increasing drug concentration, this effect became less prominent (see Figure 7b). Once internalized via endocytosis, the PTX-loaded nanocarriers swiftly and efficiently released their cargo when the enzymes and acidic conditions in endosomes triggered the cleavage of the pH-sensitive acyclic ketal bond [43,44]. Drug-free MPEO_44_-*b*-PCL_17_ NPs were also tested up to the applied maximal concentration (0.67 mg·mL^−1^) with no significant cytotoxic activity (Appendix A). Last but not least, the negligible toxicity of the unloaded-MPEO_44_-*b*-PCL_17_ NPs emphasized that the presented nanocarrier system produced no toxic degradation products, and at any rate, the products (PCL and PEO) are well-known and FDA-approved as environmentally friendly blocks.

## 4. Conclusions

In summary, well-defined nanoparticles prepared by the self-assembly of the new amphiphilic MPEO_44_-*b*-PCL_17_ block copolymer in an aqueous solution were presented. The NPs structure was characterized in detail by DLS, SLS, SAXS, and TEM. On decreasing pH the acid-labile ketal linker enabled the disassembly of the nanoparticles in a buffer that simulated the acidic environment in endosomal and lysosomal compartments. As a result, the chemotherapeutic paclitaxel was released, and the polymer particles disintegrated into neutral degradation products as confirmed by SEC, ^13^C NMR, and by in vitro cell viability tests, as well. In addition, the in vitro cell viability experiments demonstrated the great potential of the pH-triggered NPs as a drug-delivery system in cancer therapy; the in vitro cytotoxicity studies showed an important increase in activity of the NP-loaded with drug and the free-drug NPs are degraded into well-known, and FDA-approved by-products and itself introduced no toxicity to cells. The particle’s hydrophilic surface coat and size below the cut-off size of the leaky pathological vasculature (NPs < 100 nm) predetermined the NPs for long circulation and efficient accumulation in solid tumors due to the EPR effect and together with the ability to release a drug at the endosomal pH with concomitant high cytotoxicity makes them suitable candidates for cancer therapy.

## Figures and Tables

**Figure 1 polymers-13-01465-f001:**
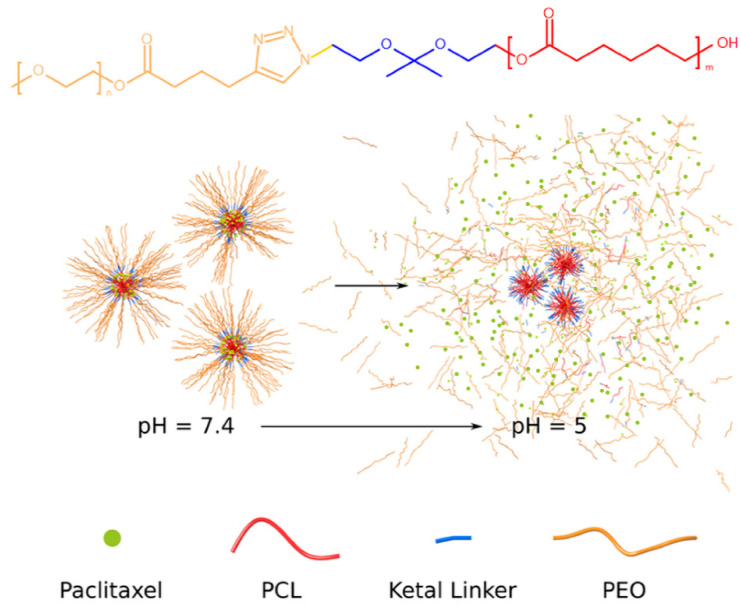
Chemical structure of the pH-triggered acid-labile block copolymer MPEO_44_-*b*-PCL_17_ (**top**) and the schematic nanoparticles assembly/disassembly mechanism at pH ~5.0 (**bottom**).

**Figure 2 polymers-13-01465-f002:**
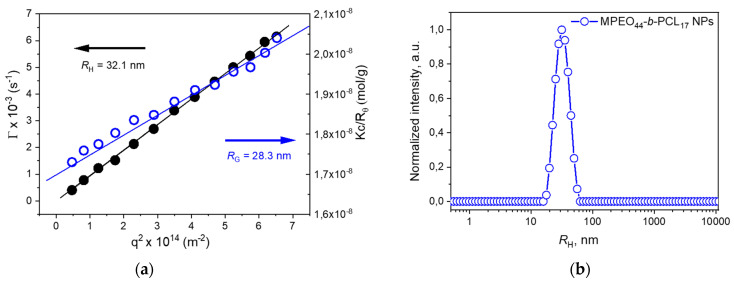
Measurement of angular dependence by DLS (•) and SLS (○) of the MPEO_44_-*b*-PCL_17_ NPs prepared using the nanoprecipitation protocol (**a**) and normalized intensity size distribution of the MPEO_44_-*b*-PCL_17_ NPs measured at angle 90° at a concentration of 1,0 mg·mL^−1^ in PBS (pH ~7.4) and at 37 °C (**b**).

**Figure 3 polymers-13-01465-f003:**
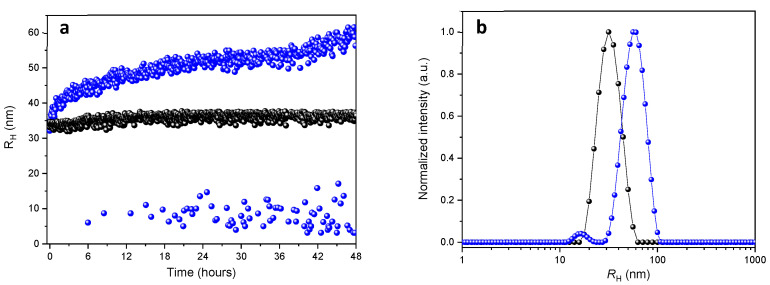
*R*_H_ evolution (**a**) and intensity distribution (**b**) measured by DLS at pH ~7.4 (•) and pH ~5.0 (•) for the MPEO_44_-*b*-PCL_17_ NPs after 48 h.

**Figure 4 polymers-13-01465-f004:**
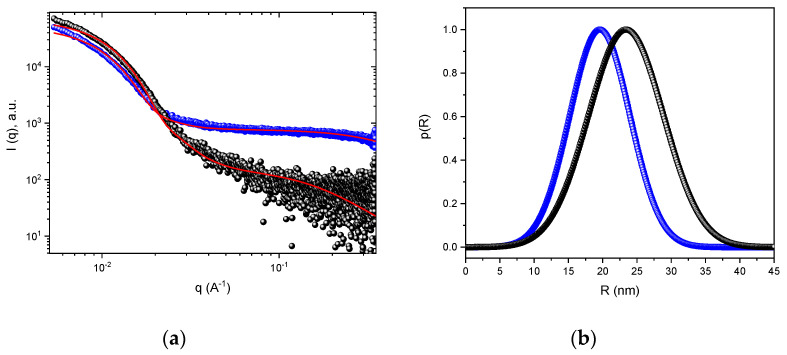
SAXS patterns of NPs after 48 h of incubation at pH ~7.4 (•) and pH ~5.0 (•) along to the fitting results—olid red lines (**a**) and the respective p(R) vs. R profiles (**b**).

**Figure 5 polymers-13-01465-f005:**
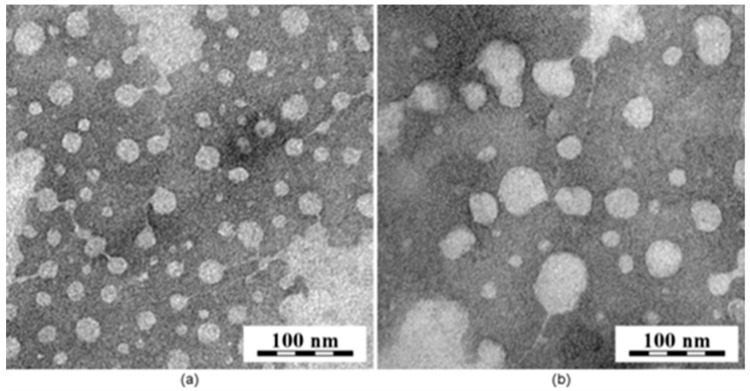
TEM images of MPEO_44_-*b*-PCL_17_ at pH ~7.4 (**a**) and pH ~5.0 (**b**) after 48 h incubation.

**Figure 6 polymers-13-01465-f006:**
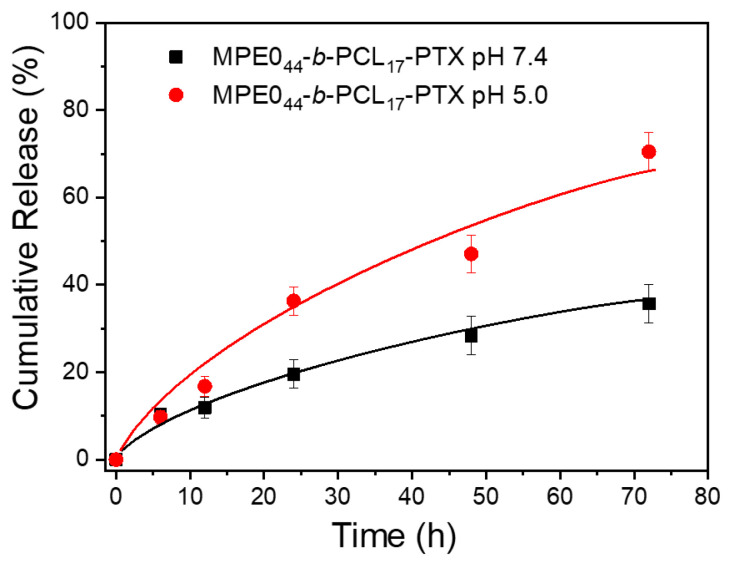
Paclitaxel release profiles from MPEO_44_-*b*-PCL_17_-PTX NPs at pH ~7.4 (▪) and pH ~5.0 (•)**.**

**Figure 7 polymers-13-01465-f007:**
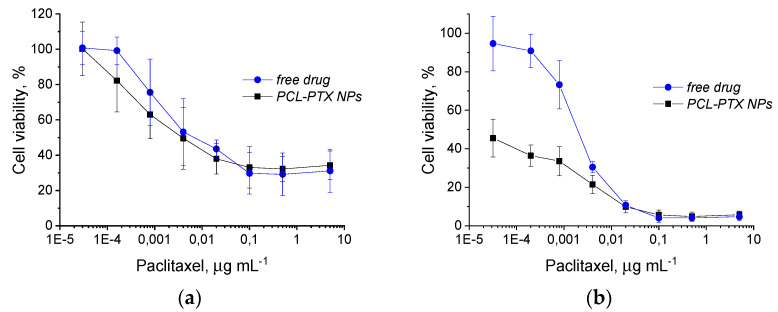
Viability of HeLa cells after 24 h (**a**) and 48 h (**b**) of incubation with different concentrations of free PTX (blue circles) and PTX-loaded MPEO_44_-*b*-PCL_17_ NPs (black squares).

**Table 1 polymers-13-01465-t001:** Macromolecular characteristics of the MPEO_44_-*b*-PCL_17_ block copolymer.

Sample	*M*_n_, ^a^ (theor.) (g moL^−1^)	*M*_n_, ^b^ (NMR) (g moL^−1^)	*M*_n_, ^c^ (SEC) (g moL^−1^)	*M*_w_/*M*_n_, ^d^ (SEC)
MPEO_44_-*b*-PCL_17_	4000	4200	3130	1.45

^a^ *M*_n_ = [M]_o_/[I]_o_ × 114 + *M*_n_ *α*-methoxy-*ω*-hydroxy-MPEO containing a ketal group (Appendix A). ^b^ *M*_n_ was calculated by ^1^H NMR spectroscopy [29]. ^c^ *M*_n_ and ^d^
*M*_w_/*M*_n_ values are relative to PS standards (Appendix A).

**Table 2 polymers-13-01465-t002:** Structural features of the prepared MPEO_44_-*b*-PCL_17_ block copolymer nanoparticles before (pH ~7.4) and after (pH ~5.0) degradation.

NPs	*R* _H_ ^a^	*R* _G_ ^b^	*R* ^c^	*Rg* _shell_ ^c^	*Rg* _gauss_ ^c^
MPEO_44-_*b*-PCL_17_ (pH ~7.4)	32.1	28.3	23.4	1.1	-
MPEO_44-_*b*-PCL_17_ (pH ~5.0)	15 and 58	-	19.5	5.6	3.0

^a^ DLS; ^b^ SLS; ^c^ SAXS; Values are given in nm.

## Data Availability

The data presented in this study are available on request from the corresponding author.

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
