# Peer review of "Development of an Acid-Labile Ketal Linked Amphiphilic Block Copolymer Nanoparticles for pH-Triggered Release of Paclitaxel"

_polymers, 2021, doi:10.3390/polym13091465_

Round 1

Reviewer 1 Report

The paper entitled “Degradation studies of an acid-labile ketal linked amphiphilic block copolymer nanoparticles for pH-triggered release of paclitaxel” by Petrova et al. deals with the preparation and characterization of a drug delivery system based on PEO-b-PCL block copolymer NPs loaded with paclitaxel. The presence of a ketal group makes this system to be pH-sensitive and therefore to be suitable for drug release in cancer therapy.

The paper is clear and the conclusions are supported by the results. However, there are quite a lot of grammar errors. Some corrections are needed in order to increase the overall quality of the paper before publication:

  1. In the abstract, the authors must provide firstly the full name of block copolymer and then to use the abbreviation.
  2. The introduction section must be completed with several new references concerning the preparation and characterization of both block copolymers and stimuli-sensitive drug-loaded systems. Some examples are: https://doi.org/10.1016/j.progpolymsci.2017.06.001; https://doi.org/10.3390/polym10010062; https://doi.org/10.1016/j.jcis.2012.12.058; https://doi.org/10.3390/polym12071450; https://doi.org/10.1002/app.45313; https://doi.org/10.3390/polym12051018; https://doi.org/10.3390/polym13030477.
  3. Section 2.4. Why the authors have used preheated acetone and PBS solutions?! At this point, no purification was carried out?! What happens with unloaded drug, LE being around 92 %?!
  4. The equation for LC is given twice and no equation for LE was provided.
  5. Line 174: the term “micelles” is the first and the only time when it is used in this paper. If the authors consider that the block copolymer self-assembly into micelles then the term “nanoparticles” should be replaced in the entire paper with the term “micelles”.
  6. Lines 223-224: please move to the materials section the data concerning the meaning of subscripts.
  7. Please add units in table 1
  8. If SLS was used for the characterization of this micellar solutions, then the authors must calculate an aggregation number in the absence and in the presence of PTX
  9. Lines 326-329. Please revise the sentence “according, DLS results….”
  10. Line 366-367: the sentence “the drug-loaded…” is a repetition and must be deleted.

Minor corrections:

  1. Line 33: “..block copolymers…have been extensively studied”
  2. Line 37: “these nanoassemblies consist of a compact..”
  3. Line 40: “it is of particular importance that hydrophilic…”
  4. Line 66: “(PCL) hydrophobic”
  5. Line 99: “the number-average”
  6. Line 151: :in the same manner as described above”
  7. Line 213-214: “for further synthesis of the MPEO-b-PCL..”
  8. Line 273: “which indicates that…”
  9. Line 289: “a representative..distribution…is shown..”
  10. Line 309: “chains analogous to that observed by DLS”
  11. Line 324: delete “in”

Reviewer 2 Report

Petrova and coworkers developed an acid-labile block copolymer nanoparticle for potential drug delivery. The results are quite interesting. I recommend it for publication after following issues are addressed.

  1. This study is not only about degradation. I suggest the authors change the title to a general one, such as ‘Development of an acid-labile ketal linked amphiphilic block copolymer nanoparticles for pH-triggered release of 3 paclitaxel’.
  2. Line 40-43, there are many other water-soluble polymers with high biocompatibility to be used for prolonging the blood circulation time. Several studies (10.1039/C4TB01477D; doi.org/10.1002/mabi.201400152) should be included.
  3. Line 99, ‘umber-average’ should be ‘number-average’.
  4. Line 117-142, it is better to make all the equations with a consistent form. Please make the font of these equations unbolded.
  5. Equation 4 and 5 are the same. One of them should be about ‘LE’.
  6. The author should discuss in the text why the Mn estimated from NMR is much larger than the theorical one. If the Mn (NMR)= 5400, why the formular of the block polymer is MPEO44-b-PCL17?
  7. The image resolution of figure 2 should be higher.
  8. Line 271-274, the zeta potential measurement of NPs in high salt concentration (PBS, 0.15 M) is not accurate and not recommended. How did the authors perform the zeta potential measurements?
  9. Figure 7, please remove the ‘a)’ and ‘b)’ inside the figure.
